# PPO-CMA: Proximal Policy Optimization with Covariance Matrix Adaptation

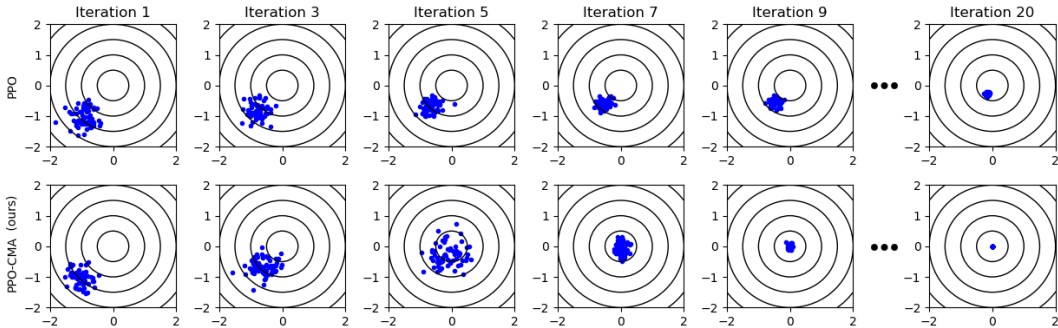

Figure 1: Comparison of Proximal Policy Optimization (PPO) and our proposed PPO-CMA method. Sampled actions $\mathbf{a} \in \mathbb{R}^2$ are shown in blue. In this "stateless" didactic example, the simple quadratic objective has the global optimum at the origin (details in Section 3.1). PPO shrinks the exploration variance prematurely, which leads to slow convergence. PPO-CMA dynamically expands the variance to speed up progress, and only shrinks the variance when close to the optimum.

## Abstract

Proximal Policy Optimization (PPO) is a highly popular model-free reinforcement learning (RL) approach. However, in continuous state and actions spaces and a Gaussian policy – common in computer animation and robotics – PPO is prone to getting stuck in local optima. In this paper, we observe a tendency of PPO to prematurely shrink the exploration variance, which naturally leads to slow progress. Motivated by this, we borrow ideas from CMA-ES, a black-box optimization method designed for intelligent adaptive Gaussian exploration, to derive PPO-CMA, a novel proximal policy optimization approach that can expand the exploration variance on objective function slopes and shrink the variance when close to the optimum. This is implemented by using separate neural networks for policy mean and variance and training the mean and variance in separate passes. Our experiments demonstrate a clear improvement over vanilla PPO in many difficult OpenAI Gym MuJoCo tasks.

## 1 Introduction

This paper proposes a new solution to the problem of policy optimization with high-dimensional continuous state and action spaces. This is a problem that has a long history in computer animation, robotics, and machine learning research. More specifically, our work falls in the domain of simulation-based Monte Carlo approaches; instead of operating with closed-form expressions of the control dynamics, we use a black-box dynamics simulator, sample actions from some distribution, simulate the results, and then adapt the sampling distribution. In recent years, such approaches have achieved remarkable success in previously intractable tasks such as real-time locomotion control of (simplified) biomechanical models of the human body (Wang et al. (2010); Geijtenbeek et al. (2013); Hämäläinen et al. (2014; 2015); Liu et al. (2016); Duan et al. (2016); Rajamäki & Hämäläinen (2017)). In 2017, Proximal Policy Optimization (PPO) provided the first demonstration of a neural

network policy that enables a simulated humanoid not only to run but also to rapidly switch direction and get up after falling (Schulman et al. (2017)). Previously, such feats had only been achieved with more computationally heavy approaches that used simulation and sampling not only during training but also in run-time (Hämäläinen et al. (2014; 2015); Rajamäki & Hämäläinen (2017)).

PPO has been quickly adopted as the default RL algorithm in popular frameworks like Unity Machine Learning Agents Juliani et al. (2018) and TensorFlow Agents (Hafner et al. (2017)). It has also been extended and applied to even more complex humanoid movement skills such as kung-fu kicks and backflips (Peng et al. (2018)). Outside the continuous control domain, it has demonstrated outstanding performance in complex multi-agent video games [1]. However, like many other reinforcement learning methods, PPO can be sensitive to hyperparameter choices and difficult to tune (Henderson et al. (2017)). In experiments by ourselves and our colleagues, we have also noticed a tendency to get stuck in local optima.

In this paper, we make the following contributions:

- We provide visualizations and evidence of how PPO's exploration variance can shrink prematurely, which leads to slow progress or getting stuck in local optima. Figure 1 illustrates this in a simple didactic example. Subsequently, we discuss how a similar exploration problem is solved in the black-box optimization domain by Covariance Matrix Adaptation Evolution Strategy (CMA-ES). CMA-ES dynamically expands the exploration variance on objective function slopes and only shrinks the variance when close to the optimum.

- We show how exploration behavior similar to CMA-ES can be achieved in RL with simple changes, resulting in our PPO-CMA algorithm visualized in Figure 1. As elaborated in Section 4, a key idea of PPO-CMA is to use separate neural networks for policy mean and variance, and train the mean and variance in separate passes. Our experiments show a significant improvement over PPO in many OpenAI Gym MuJoCo tasks (Brockman et al. (2016)).

In Appendix A, we also investigate solving the variance adaptation problem with simple variance clipping and entropy regularization. The entropy regularization was suggested by Schulman et al. (2017) but not analyzed in detail. Our results suggest that both approaches can help but they are sensitive to tuning parameters.

## 2 PRELIMINARIES

### 2.1 REINFORCEMENT LEARNING

---

**Algorithm 1** Episodic On-policy Reinforcement Learning (high-level summary)

---
1: **for** iteration=1,2,... **do**
2:     **while** iteration simulation budget $N$ not exceeded **do**
3:         Reset the simulation to a (random) initial state
4:         Run agent on policy $\pi_\theta$ for $T$ timesteps or until a terminal state
5:     **end while**
6:     Update policy parameters $\theta$ based on the observed experience $[\mathbf{s}_i, \mathbf{a}_i, r_i, \mathbf{s}'_i]$
7: **end for**

---

We consider the discounted formulation of the policy optimization problem, following the notation of Schulman et al. (2015b). At time $t$, the agent observes a state vector $\mathbf{s}_t$ and takes an action $\mathbf{a}_t \sim \pi_\theta(\mathbf{a}_t|\mathbf{s}_t)$, where $\pi_\theta$ denotes a stochastic policy parameterized by $\theta$, e.g., neural network weights. This results in observing a new state $\mathbf{s}'_t$ and receiving a scalar reward $r_t$. The goal is to find $\theta$ that maximizes the expected future-discounted sum of rewards $\mathbb{E}[\sum_{t=0}^{\infty} \gamma^t r_t]$, where $\gamma$ is a discount factor in the range $(0, 1)$. A lower $\gamma$ makes the learning prefer instant gratification instead of long-term gains.

The original PPO and the PPO-CMA proposed in this paper collect experience tuples $[\mathbf{s}_i, \mathbf{a}_i, r_i, \mathbf{s}'_i]$ by simulating a number of *episodes* in each optimization iteration. For each episode, an initial

---

[1]https://blog.openai.com/openai-five/

state $\mathbf{s_0}$ is sampled from some application-dependent stationary distribution, and the simulation is continued until a terminal (absorbing) state or a predefined maximum episode length $T$ is reached. After an iteration simulation budget $N$ is exhausted, $\theta$ is updated. This is summarized in Algorithm 1.

## 2.2 POLICY GRADIENT WITH ADVANTAGE ESTIMATION

Policy gradient methods update policy parameters by estimating the gradient $\mathbf{g} = \nabla_\theta \mathbb{E}[\sum_t^\infty \gamma^t r_t]$. There are different formulations of the gradient, of which PPO uses the following:

$$\mathbf{g} = \mathbb{E}\Big[\sum_{t=0}^{\infty} A^\pi(\mathbf{s}_t, \mathbf{a}_t)\nabla_\theta \log \pi_\theta(\mathbf{a}_t|\mathbf{s}_t)\Big], \tag{1}$$

where $A^\pi$ is the so-called *advantage function*.

Intuitively, the advantage function is positive if an explored action yields better rewards than expected. Updating $\theta$ in the direction of $\mathbf{g}$ makes negative advantage actions less likely and positive advantage actions more likely (Schulman et al. (2015b)). In practice, using modern compute graph frameworks like TensorFlow (Abadi et al. (2016)), one often does not directly operate on the gradients, but instead uses an optimizer like Adam (Kingma & Ba (2014)) to minimize the corresponding loss

$$\mathcal{L} = -\frac{1}{M}\sum_{i=1}^{M} A^\pi(\mathbf{s}_i, \mathbf{a}_i)\log \pi_\theta(\mathbf{a}_i|\mathbf{s}_i), \tag{2}$$

where $i$ denotes minibatch sample index and $M$ is minibatch size. In summary, this type of policy gradient RL simply requires a differentiable expression of $\pi_\theta$ and a way to measure $A^\pi$ for each explored state-action pair.

More specifically, the advantage function is defined as:

$$A^\pi(\mathbf{s}_t, \mathbf{a}_t) = Q^\pi(\mathbf{s}_t, \mathbf{a}_t) - V^\pi(\mathbf{s}_t). \tag{3}$$

Here, $V^\pi$ is the state value function, i.e., the expected future-discounted sum of rewards for running the agent on-policy starting from state $\mathbf{s}_t$. $Q^\pi(\mathbf{s}_t, \mathbf{a}_t)$ is the state-action value function, i.e., expected sum of rewards for taking action $\mathbf{a}_t$ in state $\mathbf{s}_t$ and then following the policy, $Q^\pi(\mathbf{s}_t, \mathbf{a}_t) = r(\mathbf{s}_t, \mathbf{a}_t) + \gamma V^\pi(\mathbf{s}_{t+1})$. Thus, the advantage can also be expressed as

$$A^\pi(\mathbf{s}_t, \mathbf{a}_t) = r(\mathbf{s}_t, \mathbf{a}_t) + \gamma V^\pi(\mathbf{s}_{t+1}) - V^\pi(\mathbf{s}_t). \tag{4}$$

In practice, $V^\pi$ is usually approximated by a critic network trained with the observed rewards summed over simulation trajectories. However, plugging such an approximation directly to Equation 4 tends to be unstable due to approximation bias. Instead, same as PPO, we use Generalized Advantage Estimation (GAE) (Schulman et al. (2015b)), which is a simple but effective way to estimate $A^\pi$ such that one can trade variance for bias.

## 3 UNDERSTANDING VARIANCE ADAPTATION IN GAUSSIAN POLICY OPTIMIZATION

This paper focuses on the case of continuous control using a Gaussian policy. In other words, the policy network outputs state-dependent mean $\mu_\theta(\mathbf{s})$ and covariance $\mathbf{C}_\theta(\mathbf{s})$ for sampling the actions. The covariance defines the exploration-exploitation balance. In practice, one often uses a diagonal covariance matrix parameterized by a vector $\mathbf{c}_\theta(\mathbf{s}) = diag(\mathbf{C}_\theta(\mathbf{s}))$. In the most simple case of isotropic unit Gaussian exploration, $\mathbf{C} = \mathbf{I}$, the loss function in Equation 2 becomes:

$$\mathcal{L} = -\frac{1}{M} \sum_{i=1}^{M} A^\pi(\mathbf{s}_i, \mathbf{a}_i) ||\mathbf{a}_i - \mu_\theta(\mathbf{s})||^2, \tag{5}$$

Following the original PPO paper, this paper uses diagonal covariance. This results in a slightly more complex loss function:

$$\mathcal{L} = -\frac{1}{M} \sum_{i=1}^{M} A^\pi(\mathbf{s}_i, \mathbf{a}_i) \sum_j [(a_{i,j} - \mu_{j;\theta}(\mathbf{s}_i))^2 / c_{j;\theta}(\mathbf{s}_i) + 0.5 \log c_{j;\theta}(\mathbf{s}_i)], \tag{6}$$

where $i$ indexes over a minibatch and $j$ indexes over action variables.

### 3.1 THE INSTABILITY CAUSED BY NEGATIVE ADVANTAGES

To gain an intuitive understanding of the training dynamics, it is important to note the following:

- Equation 5 indicates that the *policy is trained to approximate the sampled actions*, with each action weighted by $A^\pi$. Equation 5 uses an $L2$ loss, while Equation 6 uses a Bayesian loss that allows the network to model aleatoric uncertainty, expressed as the free covariance parameters (Kendall & Gal (2017)).
- As elaborated below, *actions with a negative $A^\pi$ may cause instability*, especially when one considers training for several epochs at each iteration using the same data; as pointed out by Schulman et al. (2017), this would be desirable to get the most out of the collected experience.

For negative advantages, *each minibatch gradient step drives the policy Gaussian further away from the sampled actions*. This can easily result in divergence, as shown at the top of Figure 2. In contrast, the bottom of Figure 2 shows how using only the positive advantage actions is stable; the policy Gaussian simply approximates the positive advantage action distribution. However, in this case the exploration variance shrinks prematurely, leading to poor final convergence. Note that Figure 2 shows the convergence/divergence over policy gradient iterations; Appendix D provides an illustration of gradient steps within an iteration.

Similar to Figure 1, Figure 2 depicts a "stateless" didactic example, a special case of Algorithm 1 that allows clear visualization of how the action distribution adapts. We set $\gamma = 0$, which simplifies the policy optimization objective $\mathbb{E}[\sum_{t=0}^{\infty} \gamma^t r_t] = \mathbb{E}[r_0] = \mathbb{E}[r(\mathbf{s}, \mathbf{a})]$, where $\mathbf{s}, \mathbf{a}$ denote the first state and action of an episode. Further, we use a state-agnostic $r(\mathbf{a}) = -\mathbf{a}^T \mathbf{a}$. Thus, $Q^\pi(\mathbf{s}, \mathbf{a}) = -\mathbf{a}^T \mathbf{a}$, $V^\pi(\mathbf{s}) = V^\pi = \mathbb{E}[r(\mathbf{a})] \approx \frac{1}{N} \sum_{i=0}^{N-1} r(\mathbf{a}_i)$, where $i$ is episode index, and we can compute $A^\pi$ directly using Equation 3. As everything is agnostic of agent state, a simulator is not even needed and one can feed an arbitrary constant input to a policy network. Equivalently, one can simply replace the policy network with optimized variables for the mean and variance of the action distribution.

### 3.2 PROXIMAL POLICY OPTIMIZATION: VARIANCE ADAPTATION PROBLEM

The basic idea of PPO is that one performs minibatch policy gradient updates for several epochs on the data from each iteration of Algorithm 1, while limiting changes to the policy such that it stays in the proximity of the sampled actions (Schulman et al. (2017)). PPO is a simplification of Trust Region Policy Optimization (TRPO) (Schulman et al. (2015a)), which uses a more computationally expensive approach to achieve the same.

The original PPO paper proposes two variants: 1) using a loss function that penalizes KL-divergence between the old and updated policies, and 2) using the so-called clipped surrogate loss function that limits the likelihood ratio of old and updated policies $\pi_\theta(\mathbf{a}_i|\mathbf{s}_i)/\pi_{old}(\mathbf{a}_i|\mathbf{s}_i)$. In extensive testing, Schulman et al. (2017) concluded that the clipped surrogate loss with the clipping hyperparameter $\epsilon = 0.2$ is the recommended choice. This is also the version that we use in this paper in all PPO vs. PPO-CMA comparisons.

Comparing Figure 1 and Figure 2 shows that PPO is more stable than vanilla policy gradient, but can result in somewhat similar premature shrinkage of exploration variance as in the case of only using

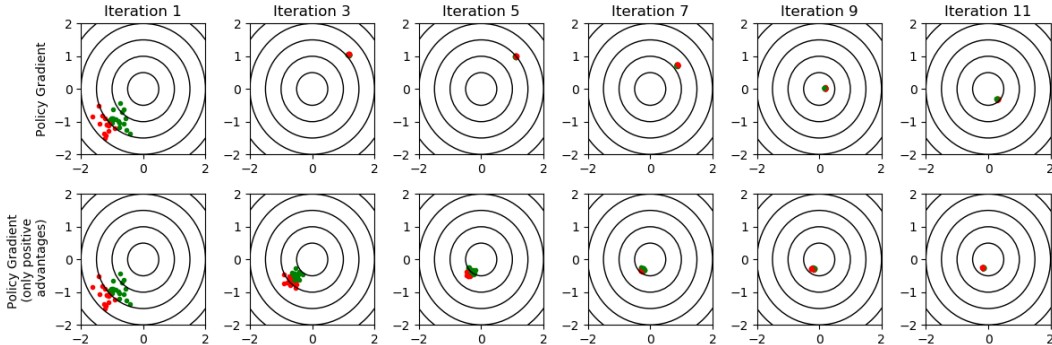

Figure 2: Comparing policy gradient variants in the didactic example of Figure 1, when doing multiple minibatch updates with the data from each iteration. Actions with positive advantage estimates are shown in green, and negative advantages in red. Top: Basic policy gradient is highly unstable. Bottom: Using only positive advantage actions, policy gradient is stable but converges prematurely.

the positive advantages for policy gradient. Although Schulman et al. (2017) demonstrated good results in MuJoCo problems with a Gaussian policy, the most impressive Roboschool results did not adapt the variance through gradient updates. Instead, the policy network only output the Gaussian mean and a linearly decaying variance with manually tuned decay rate was used.

### 3.3 Covariance Matrix Adaptation Evolution Strategy

The Evolution Strategies (ES) community has worked on similar variance adaptation and Gaussian exploration problems for decades, culminating in the widely used CMA-ES optimization method and its recent variants (Hansen & Ostermeier (2001); Hansen (2006); Beyer & Sendhoff (2017); Loshchilov et al. (2017)). CMA-ES is a black-box optimization method for finding a parameter vector $\mathbf{x}$ that maximizes some objective or fitness function $f(\mathbf{x})$. The CMA-ES core iteration is summarized in Algorithm 2.

---

**Algorithm 2** High-level summary of CMA-ES

1: **for** iteration=1,2,... **do**
2:     Draw samples $\mathbf{x}_i \sim \mathcal{N}(\boldsymbol{\mu}, \mathbf{C})$.
3:     Evaluate $f(\mathbf{x}_i)$
4:     Sort the samples based on $f(\mathbf{x}_i)$ and compute weights $\mathbf{w}_i$ based on the ranks such that best samples have highest weights.
5:     Update $\boldsymbol{\mu}$ and $\mathbf{C}$ using the samples and weights.
6: **end for**

---

Using the default CMA-ES parameters, the weights of the worst 50% of samples are set to 0, i.e., those samples are pruned and have no effect. The mean $\boldsymbol{\mu}$ is updated as a weighted average of the samples, but the covariance update is more involved. Although there is no convergence quarantee, CMA-ES performs remarkably well on multimodal and/or noisy functions such as Rastrigin if using enough samples per iteration (Hansen & Kern (2004)). For full details of the update process, the reader is referred to Hansen's excellent tutorial (Hansen (2016)).

Usually, CMA-ES and other ES variants are applied to policy optimization in the form of *neuroevolution*, i.e., directly optimizing the policy network parameters, $\mathbf{x} \equiv \theta$, with $f(\mathbf{x})$ evaluated as the sum of rewards over one or more simulation episodes (Wang et al. (2010); Geijtenbeek et al. (2013); Such et al. (2017)). This is both a benefit and a drawback; neuroevolution is simple to implement and requires no critic network, but on the other hand, the sum of rewards may not be very informative in guiding the optimization. Especially in long episodes, some explored actions may be good and should be learned, but the sum may still be low. In this paper, we are interested in whether ideas from CMA-ES could improve the sampling of actions in RL, using $\mathbf{x} \equiv \mathbf{a}$.

Instead of a single action optimization task, RL is in effect solving multiple action optimization tasks in parallel, one for each possible state. The accumulation of rewards over time further complicates matters. However, we can directly apply CMA-ES to our "stateless" didactic example with $f(\mathbf{x}) \equiv r(\mathbf{a})$, illustrated in Figure 3. Comparing this to Figure 2 reveals two key insights:

- CMA-ES prunes samples based on sorted fitness values and discards the worst half. This is visually and conceptually similar to computing the policy gradient only using positive advantages, i.e., *pruning action samples based on advantage sign.* Thus, the advantage function can be thought analogous to the fitness function, although in policy optimization with a continuous state space, one can't directly enumerate, sort, and prune the actions and advantages for each state.

- Unlike PPO and policy gradient with only positive advantages, CMA-ES avoids the premature variance shrinkage problem. Instead, it increases the variance on objective function slopes to speed up progress and shrinks the variance once the optimum has been found. This can be thought as a form of momentum that acts indirectly through the exploration variance.

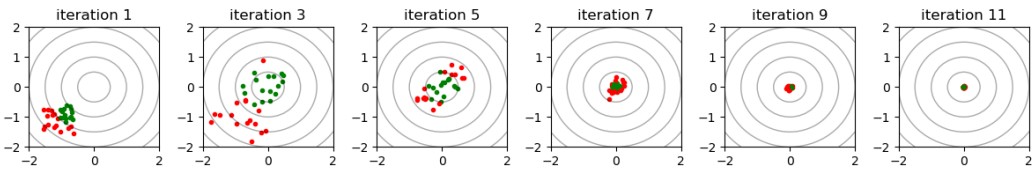

Figure 3: Our didactic example solved by CMA-ES using $\mathbf{x} \equiv \mathbf{a}$ and $f(\mathbf{x}) \equiv r(\mathbf{a})$. Red denotes pruned samples that have zero weights in updating the exploration distribution. CMA-ES expands the exploration variance while progressing on an objective function slope, and shrinks the variance when reaching the optimum.

Considering the above, a good policy optimization approach might be to only use actions with positive advantages, if one could just borrow the variance adaptation techniques of CMA-ES. In the next section, we show that this is indeed possible, resulting in our proposed PPO-CMA algorithm.

## 4   PPO-CMA

Our proposed PPO-CMA algorithm is summarized in Algorithm 3. Source code is available at GitHub[2]. PPO-CMA is simple to implement, only requiring the following minor changes to PPO:

- We use the standard policy gradient loss in Equation 6 and train only on actions with positive advantage estimates. This means that 1) we implement CMA-ES -style pruning of actions, but based on advantage sign instead of sorted fitness function values, and 2) similar to CMA-ES, the updated exploration Gaussian mean equals a weighted mean of the actions; in CMA-ES, the weights are based on the fitness sorting, whereas we use the advantages as the weights. The updated Gaussian variance is also affected by the two features below:

- We use separate neural networks for policy mean and variance, trained in separate passes. This implements the CMA-ES two-step update, as elaborated in Section 4.1.

- We maintain a history of training data over $H$ iterations, used for training the variance network. This approximates the CMA-ES evolution path heuristic, as explained in Section 4.2.

Together, the features above result in the emergence of the CMA-ES -style variance adaptation behavior shown in Figure 1. Section 4.3 also presents an optional technique for utilizing negative advantage actions, resulting in a variant of PPO-CMA we call PPO-CMA-m.

---

[2]https://github.com/ppocma/ppocma

---

**Algorithm 3** PPO-CMA

1: **for** iteration=1,2,... **do**
2:     **while** iteration simulation budget $N$ not exceeded **do**
3:         Reset the simulation to a (random) initial state
4:         Run agent on policy $\pi_\theta$ for $T$ timesteps or until a terminal state
5:     **end while**
6:     Train critic network for $K$ epochs using the experience from the current iteration
7:     Estimate advantages $A^\pi$ using GAE (Schulman et al. (2015b))
8:     Clip negative advantages to zero, $A^\pi \leftarrow \max(A^\pi, 0)$
9:     Train policy variance for $K$ epochs using experience from past $H$ iterations and Eq. 6
10:     Train policy mean for $K$ epochs using the experience from this iteration and Eq. 6
11: **end for**

---

### 4.1 Two-step updating of mean and covariance

Superficially, the core iteration loop of CMA-ES is similar to other optimization approaches with recursive sampling and distribution fitting such as the Cross-Entropy Method (De Boer et al. (2005)) and Estimation of Multivariate Normal Algorithm (EMNA) (Larrañaga & Lozano (2001)). However, there is a crucial difference: in the so-called Rank-$\mu$ update, *CMA-ES first updates the covariance and only then updates the mean* (Hansen (2016)). This has the effect of elongating the exploration distribution along the best search directions instead of shrinking the variance prematurely, as shown in Figure 4. This has also been shown to correspond to a natural gradient update of the exploration distribution (Ollivier et al. (2017)).

In PPO-CMA, we implement the two-step update by using separate neural networks for the mean and variance. We first train the variance network while keeping the mean fixed, and then vice versa, using the Gaussian policy gradient loss in Equation 6.

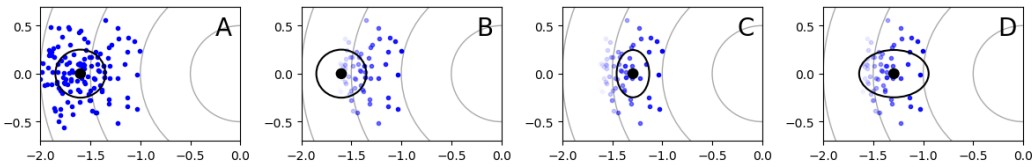

Figure 4: The difference between joint and separate updating of mean and covariance, denoted by the black dot and ellipse. A) sampling, B) pruning and weighting of samples based on fitness, C) EMNA-style update, i.e., estimating mean and covariance based on weighted samples, D) CMA-ES update, where covariance is estimated before updating the mean.

### 4.2 Evolution path

CMA-ES also features the so-called evolution path heuristic, where a component $\alpha \mathbf{p}^{(i)} \mathbf{p}^{(i)T}$ is added to the covariance, where $\alpha$ is a scalar, the $(i)$ superscript denotes iteration index, and $\mathbf{p}$ is the evolution path (Hansen (2016)):

$$\mathbf{p}^{(i)} = \beta_0 \mathbf{p}^{(i-1)} + \beta_1 (\boldsymbol{\mu}^{(i)} - \boldsymbol{\mu}^{(i-1)}). \tag{7}$$

Although the exact computation of the default $\beta_0$ and $\beta_1$ multipliers is rather involved, Equation 7 essentially amounts to first-order low-pass filtering of the steps taken by the distribution mean between iterations. When CMA-ES progresses along a continuous slope of the fitness landscape, $||\mathbf{p}||$ is large, and the covariance is elongated and exploration is increased along the progress direction. Near convergence, when CMA-ES zigzags around the optimum in a random walk, $||\mathbf{p}|| \approx 0$ and the evolution path heuristic has no effect.

PPO-CMA approximates the evolution path heuristic by keeping a history of $H$ iterations of data and sampling the variance training minibatches from the history instead of only the latest data. Same

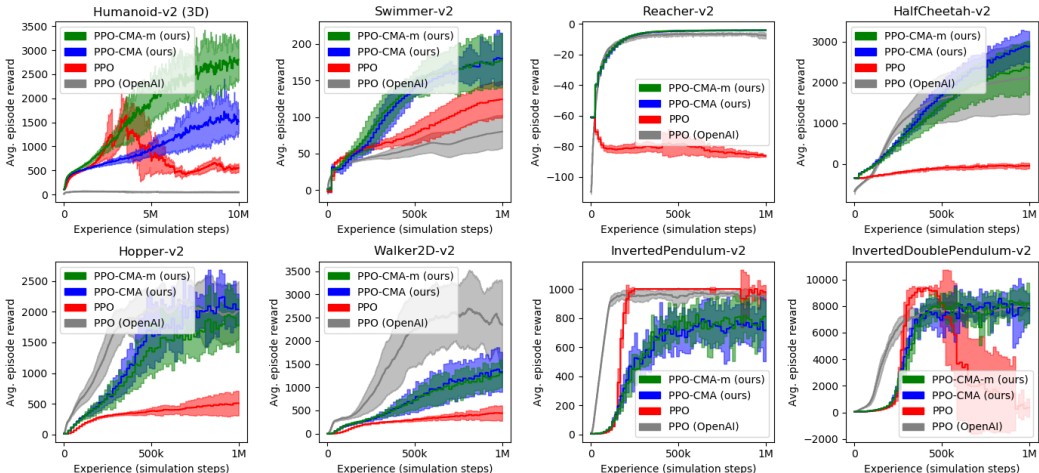

Figure 5: Means and standard deviations of training curves in OpenAI Gym MuJoCo tasks. The humanoid results are from 5 runs and others from 10 runs. Red denotes our vanilla PPO implementation with the same hyperparameters as PPO-CMA and PPO-CMA-m, providing a controlled comparison of the effect of algorithm changes. Gray denotes OpenAI's baseline PPO implementation using their default hyperparameters and training scripts for these MuJoCo tasks.

as the original evolution path heuristic, this elongates the variance for a given state if the mean is moving in a consistent direction. We do not implement the CMA-ES evolution path heuristic directly, because this would need yet another neural network to maintain and approximate a state-dependent $\mathbf{p}(\mathbf{s})$. Similar to exploration mean and variance, $\mathbf{p}$ is a CMA-ES algorithm state variable; in policy optimization, such variables become functions of agent state and need to be encoded as neural network weights.

Appendix B provides an empirical analysis of the effect of different $H$ values. Values larger than 1 minimize the probability of low-reward outliers, and PPO-CMA does not appear to be highly sensitive to the exact choice.

### 4.3 LEVERAGING NEGATIVE ADVANTAGES: PPO-CMA-M

Disregarding negative-advantage actions may potentially discard valuable information. We observe that assuming linearity of advantage around the current policy mean $\boldsymbol{\mu}(\mathbf{s}_i)$, it is possible to mirror negative-advantage actions about the mean to convert them to positive-advantage actions. More precisely, we set $\mathbf{a}_i' = 2\boldsymbol{\mu}(\mathbf{s}_i) - \mathbf{a}_i, A^\pi(\mathbf{a}_i') = -A^\pi(\mathbf{a}_i)\psi(\mathbf{a}_i, \mathbf{s}_i)$, where $\psi(\mathbf{a}_i, \mathbf{s}_i)$ is a Gaussian kernel (we use the same shape as the policy) that assigns less weight to actions far from the mean. We call this PPO-CMA variant PPO-CMA-m. The mirroring drives the policy Gaussian away from worse than average actions, but in a way that does not diverge (see Appendix D for details). If the linearity assumption holds, the mirroring effectively doubles the available experience.

In the CMA-ES literature, a related technique is to use a negative covariance matrix update procedure (Jastrebski & Arnold (2006); Hansen & Ros (2010)).

## 5 EVALUATION

We evaluate PPO-CMA using the 7-task MuJoCo-1M benchmark used by Schulman et al. (2017) plus the more difficult 3D humanoid locomotion task. We performed the following experiments:

- The green, blue, and red curves in Figure 5 visualize the results using PPO-CMA, PPO-CMA-m, and PPO with exactly same implementation and hyperparameters. This ensures that the differences are due to algorithm changes instead of hyperparameters. PPO-CMA and PPO-CMA-m perform better than PPO in 7 out of 8 tasks. The hyperparameters are

> detailed in Appendix B. PPO-CMA and PPO-CMA-m yield roughly similar results, except for the humanoid task where PPO-CMA-m is clearly superior.

- To test whether our results generalize to other hyperparameter settings, we generated 50 random hyperparameter combinations as detailed in Appendix C, and used them to train the 2D walker using both PPO and PPO-CMA. After 1M simulation steps, PPO-CMA and PPO achieved mean episode reward of 393 and 161, respectively. The standard deviations were 208 and 81. A Welch t-test indicates that the difference is statistically significant ($p < 0.001$). Although the individual training runs have high reward variance, PPO-CMA wins in 47 out of 50 runs.

- Reinforcement learning algorithms are notoriously sensitive to implementation details, and even closely related algorithms might have different optimal parameters. Thus, to compare our results against a finetuned PPO implementation, Figure 5 also shows the performance of OpenAI's baseline PPO [3]. In this case, PPO-CMA produces better results than PPO in 4 out of 8 tasks.

Overall, PPO-CMA and PPO-CMA-m often progress somewhat slower than PPO, but appear less likely to diverge or stop improving.

Appendix A provides further comparisons that also visualize the variance adaptation behavior. To augment the visual comparison, Appendix C presents statistical significance testing results.

## 6 RELATED WORK

In addition to PPO, our work is closely related to Continuous Actor Critic Learning Automaton (CACLA) (van Hasselt & Wiering (2007)). Similar to PPO-CMA, CACLA uses the sign of the advantage estimate – in their case the TD-residual – in the updates, shifting policy mean towards actions with positive sign. The paper also observes that using actions with negative advantages can have an adverse effect. In light of our discussion of how only using positive advantage actions guarantees that the policy stays in the proximity of the collected experience, CACLA can be viewed as an early Proximal Policy Optimization approach, which we extend with CMA-ES style variance adaptation.

Although PPO is based on a traditional policy gradient formulation, there is a line of research suggesting that the so-called natural gradient can be more efficient in optimization (Amari (1998); Wierstra et al. (2008); Ollivier et al. (2017)). Through the connection between CMA-ES and natural gradient, PPO-CMA is related to various natural gradient RL methods (Kakade (2002); Peters & Schaal (2008); Wu et al. (2017)), although the evolution path heuristic is not motivated from the natural gradient perspective (Ollivier et al. (2017)).

PPO represents on-policy RL methods, i.e., experience is assumed to be collected on-policy and thus must be discarded after the policy is updated. Theoretically, off-policy RL should allow better sample efficiency through the reuse of old experience, often implemented using an experience replay buffer, introduced by Lin (1993) and recently brought back to fashion (e.g, Mnih et al. (2015); Lillicrap et al. (2015); Schaul et al. (2015); Wang et al. (2016)). PPO-CMA can be considered as a hybrid method, since the policy mean is updated using on-policy experience, but the history or replay buffer for the variance update also includes older off-policy experience.

In addition to neuroevolution (discussed in Section 3.3), CMA-ES has been applied to continuous control in the form of trajectory optimization. In this case, one searches for a sequence of optimal controls given an initial state, and CMA-ES and other sampling-based approaches (Al Borno et al. (2013); Hämäläinen et al. (2014; 2015); Liu et al. (2016)) complement variants of Differential Dynamic Programming, where the optimization utilizes gradient information (Tassa et al. (2012; 2014)). Although trajectory optimization approaches have demonstrated impressive results with complex humanoid characters, they require more computing resources in run-time.

Finally, it should be noted that PPO-CMA falls in the domain of model-free reinforcement learning approaches. In contrast, there are several model-based methods that learn approximate models

---

[3] `https://github.com/openai/baselines`, we use the default parameters of run_mujoco.py, run_humanoid.py

of the simulation dynamics and use the models for policy optimization, potentially requiring less simulated or real experience. Both ES and RL approaches can be used for the optimization (Chatzilygeroudis et al. (2018)). Model-based algorithms are an active area of research, with recent work demonstrating excellent results in limited MuJoCo benchmarks (Chua et al. (2018)), but model-free approaches still dominate the most complex continuous problems such as humanoid movement.

For a more in-depth review of continuous control policy optimization methods the reader is referred to Sigaud & Stulp (2018) or the older but mathematically more detailed Deisenroth et al. (2013).

## 7  CONCLUSION

Proximal Policy Optimization (PPO) is a simple, powerful, and widely used model-free reinforcement learning approach. However, we have shown that in continous control with Gaussian policy, PPO can adapt the exploration variance in an unreliable manner, which explains how PPO may converge slowly or get stuck in local optima.

As a solution to the variance adaptation problem, we have proposed the PPO-CMA algorithm that implements two-step update and evolution path heuristics inspired by the CMA-ES black-box optimization method. This results in an improved PPO version that is simple to implement but beats the original method in many MuJoCo tasks and is less prone to getting stuck in local optima.

Additionally, our simulations in Appendix A show that PPO's premature convergence can also be prevented with simple clipping of the policy variance or using the entropy loss term proposed by Schulman et al. (2017). However, the clipping limit or entropy loss weight needs delicate finetuning and both techniques also result in a more noisy final policy.

On a more general level, one can draw the following conclusions and algorithm design insights from our work:

- We provide a new link between RL and ES approaches to policy optimization. Typically, ES is used for policy optimization in the form of neuroevolution, i.e., directly sampling the neural network weights. In contrast, we demonstrate how CMA-ES can be used to sample actions, but such that the sampling Gaussian is conditional on agent state, implemented using the policy network. Essentially, multiple parallel CMA-ES optimizations of actions are done for different agent states, and the neural networks store and interpolate algorithm state – exploration mean and variance – as a function of agent state. This is enabled by treating the advantage function as the fitness function, and approximating the CMA-ES sorting and pruning operations by clipping advantage values below a limit to zero. Sorting actions based on advantage or fitness is not possible because with a continuous state space, one can't enumerate all the actions sampled for a given state.

- Our work highlights how gradient-based RL can have problems if the gradient affects exploration in subsequent iterations, which is the case with Gaussian policies. The fundamental problem is that a gradient that causes an increase in the expected rewards does not quarantee further increases in subsequent iterations. Instead, one should adapt exploration such that it provides good results over the whole training process. We have shown that one way to achieve this can be through an approximation of CMA-ES variance adaptation.

- To understand the differences, similarities, and problems of policy optimization methods, it can be useful to visualize "stateless" special cases such as the one in Figure 1. PPO's problems were not at all clear to us until we created the visualizations, originally meant for teaching.

Thinking of the advantage function not as a way to compute accurate gradient but as a tool for pruning the learned actions also begs the question of what other pruning methods might be applicable. Presently, we are experimenting with continuous control Monte Carlo Tree Search methods (e.g., Rajamäki & Hämäläinen (2018)) for better exploration and pruning.

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

## A  FURTHER ANALYSIS OF PPO'S VARIANCE ADAPTATION

### A.1  VARIANCE ADAPTATION IN HOPPER-V2

To further inspect the variance adaptation problem, we use the Hopper-V2 MuJoCo environment, where the monopedal agent gets rewards for traveling forward while keeping the upper part of the body upright. There is a local optimum that tends to attract policy optimization: Instead of discovering a stable gait, the agent may greedily lunge forward and fall, either right at episode start or after one or a few steps.

Figure 6 shows the training trajectories from 20 independent runs with different random seeds. The figure shows both the growth of rewards and adaptation of variance, the latter plotted as the policy standard deviation averaged over all episodes and explored actions of each iteration. The figure also includes results of the same using our proposed PPO-CMA algorithm with the same hyperparameters.

Figure 6 uses red color to denote the training runs where the policy failed to escape the local optimum. Looking at the reward and variance curves together, one sees that PPO produces worse results than PPO-CMA and PPO's average standard deviation decreases faster. Some of PPO's failure cases also exhibit erratic sharp changes in standard deviation. PPO-CMA has no clear low-reward outliers adapts variance robustly and consistently.

Many PPO implementations, including OpenAI's baseline, use some form of automatic normalization of state observations, as some MuJoCo environments have observation variables with both very large and very small scales. Figure 7 shows 20 Hopper training curves with the same normalization we use in our evaluation in Section 5. PPO-CMA is more robust to the normalization.

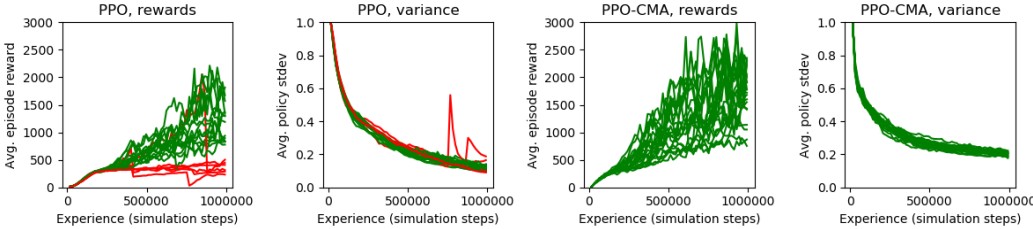

Figure 6: Comparing PPO and PPO-CMA in 20 training runs with the Hopper-V2 environment. Red trajectories denote clear failure cases where the reward plateaus and the agent only falls forward or takes one or a few steps. PPO's variance decreases faster and there is also some instability with variance peaking suddenly, associated with a decrease of rewards. PPO-CMA has no similar plateaus and adapts variance robustly and consistently.

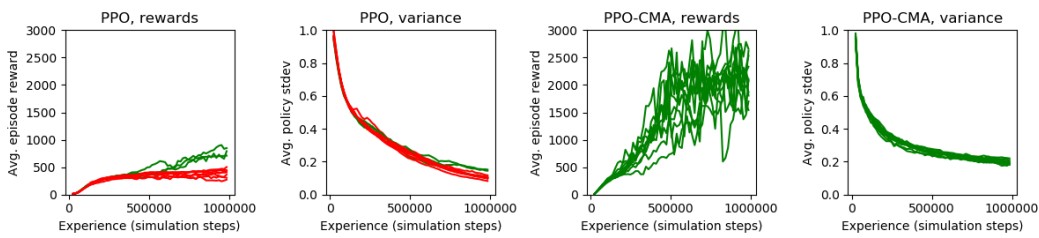

Figure 7: Same as Figure 6 but with the automatic state observation normalization described in Appendix B. PPO-CMA is more robust to the normalization. PPO failure cases are associated with lower variance.

### A.2  CLIPPING AND INITIALIZING THE POLICY NETWORK

Similar to previous work, we use a fully connected policy network with a linear output layer and treat the variance output as log variance $\mathbf{v} = \log(\mathbf{c})$. In our initial tests with PPO, we ran into numerical precision errors which could be prevented by soft-clipping the mean as $\boldsymbol{\mu}_{clipped} = \mathbf{a}_{min} + (\mathbf{a}_{max} - \mathbf{a}_{min}) \otimes \sigma(\boldsymbol{\mu})$, where $\mathbf{a}_{max}$ and $\mathbf{a}_{min}$ are the action space limits. Similarly, we clip the log variance as $\mathbf{v}_{clipped} = \mathbf{v}_{min} + (\mathbf{v}_{max} - \mathbf{v}_{min}) \otimes \sigma(\mathbf{v})$, where $\mathbf{v}_{min}$ is a lower limit parameter, and $\mathbf{v}_{max} = 2\log(\mathbf{a}_{max} - \mathbf{a}_{min})$.

To ensure a good initialization, we pretrain the policy in supervised manner with randomly sampled observation vectors and a fixed target output $\mathbf{v}_{clipped} = 2\log(0.5(\mathbf{a}_{max} - \mathbf{a}_{min}))$ and $\boldsymbol{\mu}_{clipped} = 0.5(\mathbf{a}_{max} + \mathbf{a}_{min})$. The rationale behind this choice is that the initial exploration Gaussian should cover the whole action space but the variance should be lower than the upper clipping limit to prevent zero gradients. Without the pretraining, nothing quarantees sufficient exploration for all observed states.

One might think that premature convergence could be prevented simply by increasing the lower clipping limit. However, Figure 8 shows how a fairly high lower limit – considering the valid action

range (-1,1) – is needed for the policy's standard deviation in order to ensure that all training runs escape the local optimum. This is not ideal, as it causes considerable motor noise that does not vanish as training progresses. Excessive motor noise is undesireable especially in animation and robotics applications. Finetuning the limit is also tedious, as to large values rapidly lead to worse results.

## A.3 ENTROPY LOSS

The original PPO paper (Schulman et al. (2017)) discusses adding an entropy loss term to penalize low variance and prevent premature convergence, although their recommended settings for continuous control tasks do not use the entropy loss and the effect of the loss weight was not empirically investigated. Figure 8 shows how the entropy loss works similarly to the lower clipping limit, i.e., increasing the entropy loss weight helps to mimimize getting stuck in the local optimum. Too large values cause a rapid decrease in average performance and the entropy loss also results in a more noisy final policy. To mitigate this, some PPO implementations such as the one in Unity Machine Learning Agents framework anneal the entropy loss weight to zero during training; however, this adds the cost of finetuning even more hyperparameters. Ideally, one would like the variance adaptation to be both efficient and automatic.

If one wants to use PPO with variance clipping or entropy loss instead of PPO-CMA, our recommendation is to try variance clipping, as it results in better results in Figure 8. Figure 9 also shows how the entropy loss results in less stable variance curves.

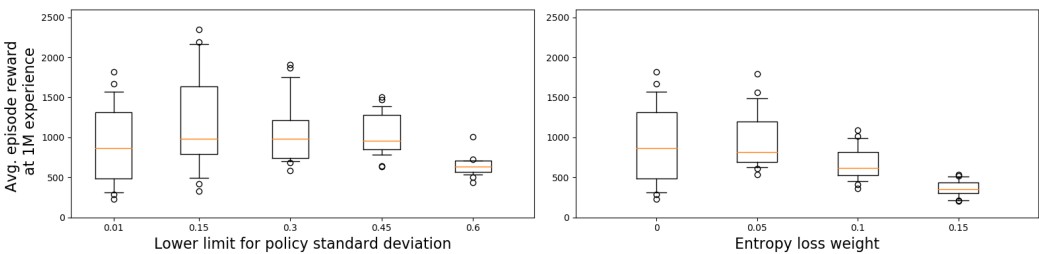

Figure 8: Boxplot of PPO results of 20 training runs of Hopper-V2 with different entropy loss weights and lower clipping limits for policy's standard deviation. The plots are from the last iteration where a limit of 1M total simulation steps was reached. The upper standard deviation limit is 1 in all plots and the entropy loss weight plots use a lower clipping limit of 0.01. Large values of either parameter can help escaping the local optimum of simply falling forward or taking only 1 or few steps (rewards below 1000), but at the same time, too large values impair average and best-case performance.

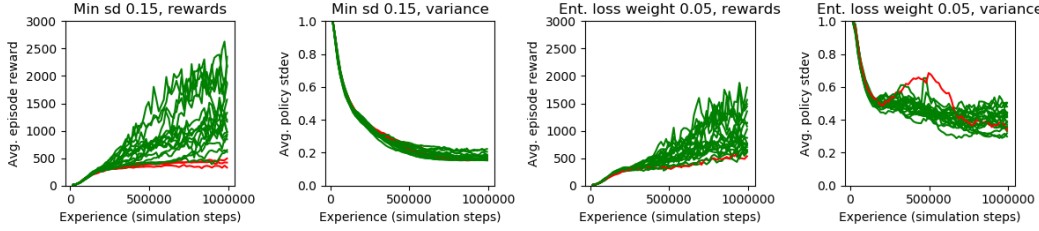

Figure 9: Comparing the effect of PPO variance clipping and entropy loss. Even with a fairly low weight, the entropy loss can lead to worse results and cause unstable increases in variance that yield low rewards (the red trajectory in the rightmost images).

## B  HYPERPARAMETERS AND IMPLEMENTATION DETAILS

This section describes details of our PPO-CMA and PPO implementations. OpenAI's baseline PPO was used in Section 5 without any modifications.

We use the policy network output clipping described in Section A.2 with a lower standard deviation limit of 0.01. Thus, the clipping only ensures numerical precision but has little effect on convergence, as illustrated in Figure 8. We use the clipping because without it, we ran into numerical precision problems in PPO (but not with PPO-CMA) in environments such as the Walker2d-V2. The clipping is not necessary for PPO-CMA in our experience, but we still use with both algorithms it to ensure a controlled and fair comparison. Following Schulman's original PPO code, we also use episode time as an extra feature for the critic network to help minimize the value function prediction variance arising from episode termination at the environment time limit. Note that as the feature augmentation is not done for the policy, this has no effect on the usability of the training results.

Table 1 lists all our hyperparameters.

| Hyperparameter | Value |
| --- | --- |
| Iteration simulation budget ($N$) | 16000 (32000 for humanoid) |
| Training epochs per iteration ($K$) | 20 |
| Variance history length ($H$) | 3 (7 for humanoid) |
| Minibatch size | 256 |
| Learning rate | 3e-4 |
| Network width | 128 (64 for humanoid) |
| Num. hidden layers | 2 |
| Activation function | Leaky ReLU |
| Action repeat | 2 |
| Critic loss | L1 |

Table 1: Hyperparameters used in our PPO and PPO-CMA implementation

We use the same network architecture for all neural networks. Action repeat of 2 means that the policy network is only queried for every other simulation step and the same action is used for two steps. This speeds up training. $N$ is specified in simulation steps. $H$ is used only for PPO-CMA. Figure 5 is generated assuming early termination to reduce variance, i.e., for each training run, the graphs use the best scoring iteration's results so far.

The choice of $H = 3$ gives slightly better results than $H = 7$ in environments other than the 3D humanoid. The humanoid also needs a larger $N$ to learn; the default value of 16000 results in very noisy episode rewards that quickly plateau instead of climbing steadily. This is in line with CMA-ES, which is said to be quasi-parameter-free; one mainly needs to increase the iteration sampling budget for high-dimensional and difficult optimization problems.

Figure 11 shows the effect of different $H$ in the Hopper task. Values $H > 1$ remove low-reward outliers, but PPO-CMA does not appear to be highly sensitive to the exact choice. Comparing figures 8 and 11 suggests that the variance clipping limit, entropy loss weight and history length $H$ all behave similarly in that one of them has to be large enough to produce good results. The crucial difference is that setting $H$ to a value larger than the required one does not cause a rapid decrease in performance; thus, we find $H$ easier to adjust.

We use L1 critic loss as it seems to make both PPO and PPO-CMA less sensitive to the reward scaling. For better tolerance to varying state observation scales, we use an automatic normalization scheme where observation variable $j$ is scaled by $k_j^{(i)} = min\left(k_j^{(i-1)}, 1/\left(\rho_j + \kappa\right)\right)$, where $\kappa = 0.001$ and $\rho_j$ is the root mean square of the variable over all iterations so far. This way, large observations are scaled down but the scaling does not constantly keep adapting as training progresses. OpenAI's baseline PPO normalizes observations based on running mean and standard deviation, but we have found our normalization slightly more stable.

### B.1 ON FINETUNING AND STABILITY OF PPO-CMA AND PPO

In light of our experiments, it is reasonable to ask why OpenAI's baseline PPO gives better results than our PPO implementation in most tasks, and why our own PPO implementation is unstable in the Humanoid and Double Inverted Pendulum tasks. Our understanding is that this is due to multiple implementation details and hyperparameters that regularize the baseline PPO:

- A small iteration simulation budget of $N = 2048$, considerably less than what we use. The small $N$ yields more iterations with the same total simulation budget; however, each iteration only results in a small update because of the details below that slow down the policy adaptation, in addition to the clipped surrogate loss.

- Training only for $K = 10$ epochs, which together with the small $N$ amounts to significantly less minibatch updates per iteration.

- Using heavy gradient norm clipping with a clipping threshold of 0.5.

- Using smooth and saturating `tanh` activation functions in the neural networks, which slow down learning and result in smooth function approximations without sharp discontinuities.

- Using a smaller neural network with 64 units per layer.

In addition to the above, some PPO implementations such as the one in Unity Machine Learning Agents framework use a global diagonal covariance independent of state, which adapts more slowly as a compromise over actions of all states. As a downside, the global covariance is at least theoretically less powerful because the agent may encounter both states where it should already converge and states where it should keep exploring.

In summary, PPO may need other regularizers in addition to the epsilon parameter of the clipped surrogate loss, which can make it difficult to finetune. In contrast, PPO-CMA does not seem to need such regularization. On the other hand, PPO-CMA can make large updates and a large $N$ may be needed to reduce noise of the updates in high-dimensional and/or difficult problems. This is similar to CMA-ES, where the population size is typically the main parameter to adjust. In addition, it may be beneficial to also experiment with different values for $H$ if PPO-CMA does not give good results. In such finetuning, it is useful to visualize full learning curve distributions like in Figure 11 instead of only plotting means and standard deviations, in order to see whether there are low-reward outliers where learning gets stuck in local optima. If there are, increasing $H$ may help.

## C    STATISTICAL SIGNIFICANCE TESTING DETAILS

To supplement the visual comparison of Figure 5, Table 2 compares PPO-CMA with OpenAI's baseline PPO using two-sided Welch t-tests.

| Environment | PPO-CMA | PPO (OpenAI baseline) | p-value |
|---|---|---|---|
| Hopper-v2 | 2035.1 | 2023.3 | 0.9479 |
| Walker2d-v2 | 1329.1 | **2345.3** | 0.0104 |
| HalfCheetah-v2 | **2874.7** | 2103.8 | 0.0313 |
| Reacher-v2 | **-4.1** | -7.6 | 0.0003 |
| Swimmer-v2 | **179.9** | 80.1 | 0.0000 |
| InvertedPendulum-v2 | 715.5 | **945.5** | 0.0103 |
| InvertedDoublePendulum-v2 | 7836.0 | 7838.2 | 0.9955 |
| Humanoid-v2 | **1491.6** | 49.8 | 0.0238 |

Table 2: Final mean episode rewards in the training runs of Figure 5, together with p-values from two-sided Welch t-tests. Bolded values indicate statistically significantly better results using the 0.05 significance level.

In section 5, we compare PPO-CMA and PPO on average, over a set of randomized hyperparameter combinations. The parameters were sampled uniformly in the ranges given in Table 3.

| Hyperparameter | Randomization range |
|---|---|
| Action repeat | $\{1, 2\}$ |
| Learning rate | $\left[10^{-4}, 10^{-3}\right]$ |
| Network width | $\{16, ..., 256\}$ |
| Activation function | *tanh* or *Leaky ReLU* |
| N | $\{2000, ..., 64000\}$ |
| K | $\{1, ..., 20\}$ |
| Minibatch size | $\{32, ..., 1024\}$ |

Table 3: Randomization ranges of the random hyperparameter comparison of Section 5.

# D  FURTHER VISUALIZATION OF ALGORITHM DIFFERENCES

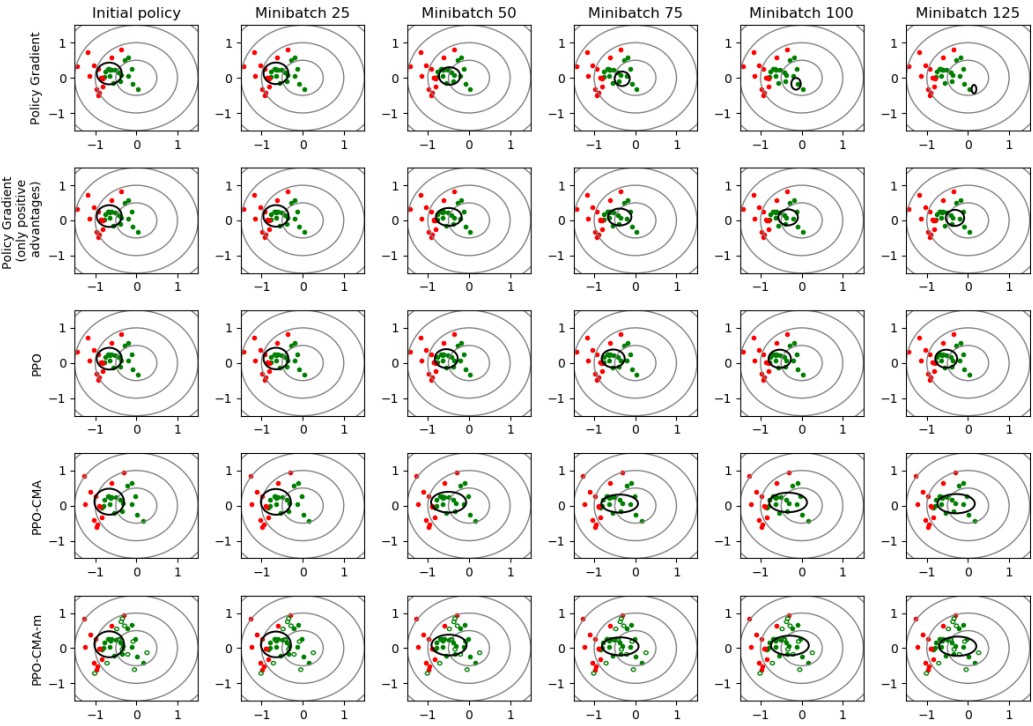

Figure 10: Algorithm differences in the didactic example of Figure 1. Here, instead of showing the results of each iteration, we show how the policy is updated in the minibatch gradient steps inside a single iteration. Actions with positive advantage estimates are shown in green, and negative advantages in red. The black ellipses denote the policy mean and standard deviation. The green non-filled circles visualize the negative-advantage actions converted to positive-advantage actions through the mirroring trick of PPO-CMA-m.

Figure 10 visualizes the effect of minibatch gradient steps taken inside one algorithm iteration in our didactic example. This complements Figures 1 and 2, which show the results of multiple iterations. The figure reveals the following:

- 1st row: Vanilla policy gradient diverges outside the data when taking multiple gradient steps with the same actions and advantages. The divergence is not caused by excessive gradient step size; each minibatch only causes a tiny change. Recall that the motivation to take multiple steps comes from PPO and TRPO; the goal is to get the most out of the collected experience.

- 2nd row: Policy gradient becomes stable when only using positive advantage actions. The policy converges to approximate the distribution of the positive advantage actions (green).

- 3rd row: PPO limits the policy change, which prevents divergence. On the other hand, it does not make as much progress as could be done based on the data.

- 4th row: PPO-CMA is stable and elongates the search distribution along the progress direction. This makes it more likely that the next iteration will generate samples close to the optimum; it lies within the unit standard deviation ellipse of the updated policy.

- 5th row: PPO-CMA-m behaves visually similar to PPO-CMA. This is in line with our other experimental results; PPO-CMA-m gives similar results as PPO-CMA in easy tasks, where the extra information of the negative advantage actions is not needed. The figure also shows how using the negative-advantage actions through the mirroring is stable; the policy converges similar to when only using positive advantages, but half of the positive-advantage actions are created through the mirroring.

## E   LIMITATIONS

We only implement an approximation of the full CMA-ES algorithm, because in addition to the mean and covariance of the search distribution, CMA-ES maintains several other auxiliary variables. In policy optimization, all algorithm state and auxiliary variables are functions of agent state and must be encoded as neural network weights, and CMA-ES has some delicately fine-tuned mechanisms which might be unstable with such approximations. We defer further investigations of this to future work, since even our simple approximation yields benefits over vanillla PPO.

We have also only focused on Gaussian policies, and the special case of diagonal covariance, i.e., the sep-CMA-ES variant (Ros & Hansen (2008)). Obviously, other exploration noise distributions such as Laplace might also work. However, Gaussian exploration is common in the literature, and the success of CMA-ES shows that it results in state-of-the-art performance in many optimization tasks beyond policy optimization.

Finally, treating the advantage function as fitness function assumes that the advantage landscape does not change significantly when policy is updated. This assumption does not hold in general for $\gamma > 0$. Thus, it is perhaps surprising that PPO-CMA works as well as it does. In future work, we plan to test whether PPO-CMA is particularly effective in problems that can be modeled using $\gamma = 0$, e.g., finding an optimal billiards shot that pockets as many balls as possible, or the humanoid climbing moves of Naderi et al. (2017), with actions parameterized as control splines instead of instantaneous torques.

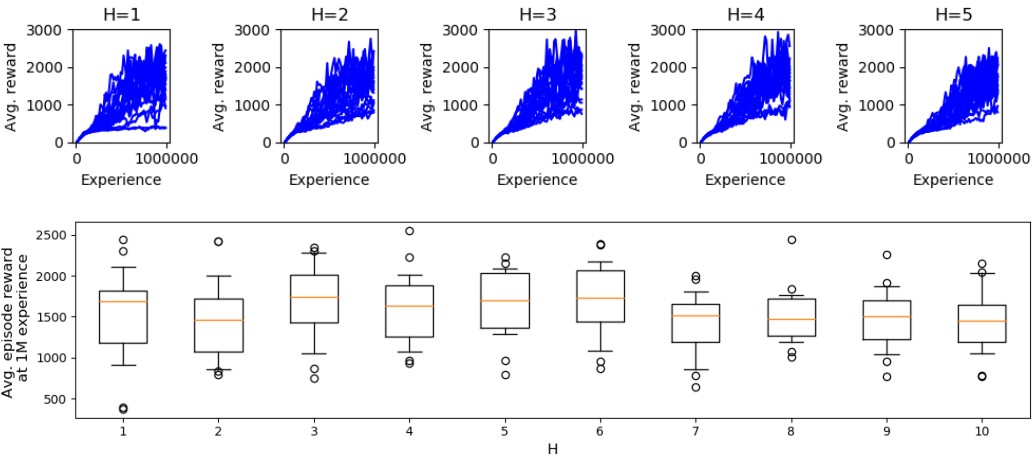

Figure 11: Training curves and final reward boxplots of PPO-CMA in 20 training runs of Hopper-V2 with different history lengths $H$. The orange lines show medians and the whiskers show the percentile range 10...90. With $H = 1$, there are outliers below a reward of 500, in which case the agent only lunges forward and falls without taking any steps.

