# OpenReview forum: "PPO-CMA: Proximal Policy Optimization with Covariance Matrix Adaptation"
_ICLR.cc/2019/Conference_

### Official Review · AnonReviewer3 · 2018-11-02
**Algorithm description is unclear**

**Rating:** 4
**Confidence:** 2

**Review:**

I have to say that this paper is not well organized. It describes the advantage function and CMA-ES, but it does not describe PPO and PPO-CMA very well. I goes through the paper twice, but I couldn't really get how the policy variance is adapted. Though the title of section 4 is "PPO-CMA", only the first paragraph is devoted to describe it and the others parts are brief introduction to CMA.

The problem of variance adaptation is not only for PPO. E.g., (Sehnke et al., Neural Networks 2009) is motivated to address this issue. They end up using directly updating the policy parameter by an algorithm like evolution strategy. In this line, algorithm of (Miyamae et al. NIPS 2010)  is similar to CMA-ES. The authors might want to compare PPO-CMA with these algorithms as baselines.

---

> ### Author Response · Authors · 2018-11-20
> **Revisions made in response to the review**
>
> Thanks for the review! We have now submitted a revised version of the paper. Below, we respond to the specific issues raised in the review.
>
>
> Related work (Sehnke et al., Miyame et al.):
>
> - Sehnke et al. and Miyamae et al. optimize by directly sampling in the space of neural network weights (neuroevolution). In contrast, PPO-CMA, like almost all RL methods, samples in the space of agent actions, which is of much smaller dimension than the space of network weights (just dozens even for humanoids), and is therefore much more amenable to optimisation by sampling. PPO-CMA improves the sampling of actions in this class of algorithms.
>
>
> Clarity of algorithm description:
>
> - We have revised the bullet point summary of PPO-CMA in the beginning of Section 4. It should now more directly highlight how CMA-ES features are implemented or approximated in PPO-CMA.
>
> - We have added Appendix D, which provides further visualization of the differences between algorithms.

---

### Official Review · AnonReviewer1 · 2018-11-03
**Very interesting paper on Covariance Matrix Adaption used to solve exploration/exploitation trade-offs in PPO**

**Rating:** 9
**Confidence:** 3

**Review:**

This is in my view a strong contribution to the field of policy gradient methods for RL in the context of continuous control. The method the authors proposed is dedicated to solving the premature convergence issue in PPO through the learning of variance control policy. The authors employ CMA-ES which is usually employed for adaptive Gaussian exploration. The method is simple and yet provides good results on a several benchmarks when compared to PPO.

One key insight developed in the paper consists in employing the advantage function as a means of filtering out samples that are associated with poorer rewards. Namely, negative advantage values imply that the corresponding samples are filtered out. Although with standard PPO such a filtering of samples leads to a premature shrinkage of the variance of the policy, CMA-ES increases the variance to enable exploration.

A key technical point is concerned with the learning of the policy variance which is cleverly done, BEFORE updating the policy mean, by exploiting a window of historical rewards over H iterations. This enables an elegant and computationally cheap means of changing the variance for a specific state.

Several experiments confirm that this method may be effective on different task when compared to PPO. Before concluding the authors carefully relate their work to prior research and delineate some limitations.

Strengths:
   o) The paper is well written.
   o) The method introduced in the paper to learn how to explore is elegant, simple and seems robust.
   o) The paper combines educational analysis through a trivial example with more realistic examples which helps the reader understand the phenomenon helping the learning as well as its practical impact.

Weaknesses:
   o) The experiments focus a lot on MuJuCo-1M. Although this task is compelling and difficult, more variety in experiments could help single out other applications where PPO-CMA helps find better control policies.

---

### Official Review · AnonReviewer2 · 2018-11-05

**Rating:** 4
**Confidence:** 4

**Review:**

This paper proposes an improvement of the PPO algorithm inspired by some components of the CMA-ES black-box optimization method. The authors evaluate the proposed method on a few Mujoco domains and compare it with PPO method using simpler exploration strategies. The results show that PPO-CMA less likely to getting stuck in local optima especially in Humanoid and Swimmer environments.

Major comments:

The reason that CMAES discards the worst batch of the solutions is that it cannot utilize the quality of the solutions, i.e., it treats every solution equally. But PPO/TRPO can surely be aware of the value of the advantage, and thus can learn to move away from the bad area. The motivation of remove the bad samples is thus not sound, as the model cannot be aware of the areas of the bad samples and can try to repetitively explore the bad area.

Please be aware that CMAES can get stuck in local optima as well. There is no general convergence guarantee of CMAES.



Detailed comments:

- In page 4, it is claimed that "actions with a negative $A^\pi$ may cause instability, especially when one considers training for several epochs at each iteration using the same data" and demonstrate this with Figure 2. This is not rigorous. If you just reduce all the negative advantage value to zero and calculate its gradient, the method is similar to just use half of step-size in policy gradient. I speculate that if you halve the step-size in "Policy Gradient" setting, the results will be similar to the "Policy Gradient(only positive advantages)" setting. Furthermore, different from importance sampling technique, pruning all the negative advantage will lose much **useful** information to improve policy. So I think this is maybe not a perfect way to avoid instability although it works in experiments.


- There have been a variety of techniques proposed to improve exploration based on derivative-free optimization method. But in my opinion, the way you combine with CMA-ES to improve exploration ability is not so reasonable. Except for the advantage function is change when policy is updated (which has mentioned in "D LIMITATIONS"), I consider that you do not make good use of the exploration feature in CMA-ES. The main reason that CMA-ES can explore better come from the randomness of parameter generation (line 2 in Algorithm 2). So it can generate more diverse policy than derivative-based approach. However, in PPO-CMA, you just replace it with the sampling of policy actions, which is not significant benefit to exploration. It more suitable to say that you "design a novel way to optimize Gaussian policy with separate network for it mean and variance inspired by the CMA-ES method rather than "provides a new link between RL and ES approaches to policy optimization" (in page 10).

- In experiments, there are still something not so clear:

1. In Figure 5, I notice that the PPO algorithm you implemented improved and then drop down quickly in Humanoid-v2 and InvertedDoublePendulum-v2, which like due to too large step-size. Have you tried to reduce it? Or there are some other reasons leading to this phenomenon.

2. What's the purpose of larger budget? You choose a bigger iteration budget than origin PPO implementation.

3. What the experiments have observed may not due to the clipping of negative reward, but could due to the scaling down of the reward. Please try reward normalization.

---

> ### Public Comment · ~Ilya_Loshchilov1 · 2018-11-07
> **Notes**
>
> "The reason that CMAES discards the worst batch of the solutions is that it cannot utilize the quality of the solutions, i.e., it treats every solution equally. "
> This comment is incorrect. First, CMA-ES does not treat every solution equally but with weights w_i thus contradicting "it treats every solution equally". Second, CMA-ES can directly utilize the quality all solutions, e.g., the weights of worst solutions are non-zero in activeCMA-ES [1] which considers all lambda solutions for the update equations.
>
> [1] "Benchmarking a Weighted Negative Covariance Matrix Update on the BBOB-2010 Noiseless Testbed" N. Hansen, R. Ros, GECCO 2010. https://hal.archives-ouvertes.fr/hal-00545728/document

---

> > ### Comment · AnonReviewer2 · 2018-12-03
> > **Irrelevant Notes**
> >
> > The comment about the original CMAES was to point out that "the motivation of remove the bad samples is thus not sound" of this paper. This comment is then confirmed by the authors and helps improve the performance.
> >
> > As for the CMAES, I didn't say "any variant of CMAES". As a heuristic search algorithm, it is certainly easy to incorporate more heuristics into CMAES. The question is then that, are the heuristics provably better? Unfortunately no answer.
> >
> > Therefore, the motivation is still not solid to me. I don't see a real difference between the exploration principles of CMAES and PPO. The advantage of PPO-CMA may vanish when controlling the exploration of PPO better.

---

> > > ### Author Response · Authors · 2018-12-03
> > > **Exploration in PPO and PPO-CMA**
> > >
> > > A note on anonymity: please note that the commenter who started this thread is not an author and is not connected to this work.
> > >
> > > “The advantage of PPO-CMA may vanish when controlling the exploration of PPO better.”
> > >
> > > In PPO, tools for controlling the exploration are limited; PPO-CMA contributes a novel exploration approach. We stay true to the PPO core idea of keeping the updated policy in the proximity of the old policy, while allowing CMA-ES style exploration where the exploration variance grows in the progress direction, as illustrated in Figure 10. This can enable larger updates in subsequent iterations, illustrated in Figure 1.
> > >
> > > In the original PPO, exploration is primarily adjusted by the epsilon parameter and the entropy loss weight. Larger epsilon values allow larger policy updates, but in practice the epsilon needs to be small to avoid similar instability as with Policy Gradient in figures 2 and 10. Larger entropy loss weight makes the algorithm prefer larger exploration variance, but too large values can easily cause worse results, as shown in figures 8 and 9. This makes finetuning the parameter tedious. As we discuss in the paper, it is also possible to design a predetermined variance annealing scheme for a specific task, as was done in the humanoid case of the original PPO paper. However, this is likely to require time-consuming trial-and-error iteration. Ideally, one would like to be able to use the same approach for all tasks.
> > >
> > > We make no claims of PPO-CMA being the ultimate or only solution to improving exploration; in future work, it should be possible to combine PPO-CMA with other approaches such as intrinsic motivation.

---

> ### Author Response · Authors · 2018-11-20
> **Revisions made in response to the review**
>
> Thanks for the review! We have now submitted a revised version of the paper. Below, we respond to the specific issues raised in the review.
>
>
> Discarding negative advantages:
>
> - We discovered a trivial modification that allows utilization of negative-advantage actions through a local linearity assumption, explained in section 4.3. The results have been added to Figure 5. The performance is considerably better in the humanoid case, and similar in the other tests.
>
>
> CMA-ES has no convergence guarantee:
>
> - We now clarify this in the 2nd paragraph of Section 3.3
>
>
> Isn't using only positive advantages the same as halving the step size in Policy Gradient?
>
> - No. The new Appendix D and Figure 10 clarify the the difference between positive and negative advantages. The figure shows that even though each minibatch gradient step only causes a tiny update, Policy Gradient with negative advantages diverges after sufficiently many steps.
>
>
>
> "In PPO-CMA, you just replace it with the sampling of policy actions, which is not significant benefit to exploration":
>
> - Both CMA-ES and PPO-CMA explore similarly, i.e., parameter vectors or actions are sampled from a Gaussian distribution. The difference is that in PPO-CMA, the Gaussian is conditioned on the agent state. This is implemented using the policy network that outputs the state-dependent mean and variance, which are used to sample the actions. We contribute by showing how CMA-ES can be approximated with this type of neural network parameterization of the action-space exploration Gaussian.
>
> - Figure 1 visualizes how this can lead to better exploration than PPO.
>
> - The added visualization in Figure 10 (Appendix D) clarifies how the policy Gaussian adapts through the minibatch updates and how PPO-CMA differs from PPO and Policy Gradient.
>
> - A limitation of our work is that we only use diagonal covariance, which corresponds to the sep-CMA-ES variant instead of full CMA-ES. We now clarify this in the limitations section.
>
>
>
> A new link between RL and ES approaches?
>
> - We now clarify this in the conclusion (first bullet-point)
>
>
>
> PPO instability in Humanoid-V2 and InvertedDoublePendulum?
>
> - Although our implementation uses the same epsilon parameter as the original PPO, the original PPO implementation has multiple other regularizers that also play a role. These are discussed in Section B.1., which we have revised for clarity.
>
> - One explanation for the instability is that there is no guarantee that overshoots like those visualized for vanilla policy gradient in Appendix D are not possible in PPO when close to convergence, in particular for tasks that are very sensitive to small policy changes. Both the Humanoid and InvertedDoublePendulum are such tasks.
>
> - The instability is one of the reasons why we also include the OpenAI baseline PPO results using their implementation and default hyperparameters. This should allow a fair comparison.
>
>
>
> Large iteration simulation budget (N)?
>
> - With our implementations, a large N seems to work better for both PPO and PPO-CMA. With a small N, the updates become very noisy. Although this could be counteracted by additional regularization, as discussed in Section B.1., we preferred to keep our implementations as simple as possible.
>
>
>
>
> Reward normalization?
>
> - We use advantage normalization based on standard deviation
>
> - We initially also tried reward normalization, but this produced slightly worse results.

---

### Author Response · Authors · 2018-11-20
**Revised paper**

We thank all the reviewers for their comments! We've now uploaded a revised pdf. A summary of revisions is included below.

NOTE: We will also respond to the reviews separately to answer questions and clarify how the paper revisions address the critique.


REVISION SUMMARY:

- Revised the bullet point algorithm summary in the beginning of Section 4. It should now better explain how CMA-ES features are implemented or approximated by PPO-CMA.

- Clarified the conclusions, in particular how PPO-CMA provides a new connection between RL and ES approaches to policy optimization

- Added Appendix D to better visualize the differences between algorithms and the different effect of positive and negative advantages.

- Added the subsection 4.3 on PPO-CMA-m (a trivial modification that allows utilizing negative-advantage actions), in response to the comments of R2. PPO-CMA-m results are now included in Figure 5. The results are considerably better than PPO-CMA in Humanoid-v2, and similar in other tests.

---

### Meta-Review · Area_Chair1 · 2018-12-13
**Improvement needed**

**Confidence:** 4
**Recommendation:** Reject

**Metareview:**

This paper proposes to improve the exploration in the PPO algorithm by applying CMA-ES. Major concerns of the paper include: paper editing can be improved; the choices of baselines used in the paper may be not reasonable; flaws in comparisons with SOTA. It is also not quite clear why CMA can improve exploration, further justification required. Overall, this paper cannot be published in its current form.